

# The control of short-term ice mélange weakening episodes on calving activity at major Greenland outlet glaciers

Adrien Wehrlé[1], Martin P. Lüthi[1], and Andreas Vieli[1]

[1]Institute of Geography, University of Zurich, 8052 Zurich, Switzerland

**Correspondence:** Adrien Wehrlé (adrien.wehrle@geo.uzh.ch)

**Abstract.** The dense mixture of iceberg of various sizes and sea ice observed in many of Greenland's fjords, called ice mélange (sikussak in Greenlandic), has been shown to have a significant impact on the dynamics of several Greenland tidewater glaciers mainly through the seasonal support it provides to the glacier terminus in winter. However, a clear understanding of shorter-term ice mélange dynamics is still lacking, mainly due to the high complexity and variability of the processes at play at the ice-ocean boundary. In this study, we use a combination of Sentinel-1 radar and Sentinel-2 optical satellite imagery to investigate in detail intraseasonal ice mélange dynamics and its link to calving activity at three major outlet glaciers: Kangerdlugssuaq Glacier, Helheim Glacier and Sermeq Kujalleq in Kangia (Jakobshavn Isbræ). In those fjords, we identified recurrent ice mélange weakening (IMW) episodes consisting in the up-fjord propagation of a discontinuity between jam-packed and weaker ice mélange towards the glacier terminus. At a late stage, i.e. when the IMW front approaches the glacier terminus, these episodes were often correlated with the occurrence of large-scale calving events. The IMW process is particularly well visible at the front of Kangerdlugssuaq glacier and presents a cyclic behavior, such that we further analyzed IMW dynamics during the June-November period from 2018 to 2021 at this location. Throughout this period, we detected 30 IMW episodes with a recurrence time of 24 days, propagating over a median distance of 5.9 km and for 17 days, resulting in a median propagation speed of 400 m/d. We found that 87% of the IMW episodes occurred prior to a calving event visible in spaceborne observations and that ~75% of all detected calving events were preceded by an IMW episode. These results therefore present the IMW process as a clear control on the calving activity of Kangerdlugssuaq glacier. Finally, using a simple numerical model for ice mélange motion, we showed that a slightly biased random motion of ice floes without fluctuating external forcing can reproduce IMW events and their cyclic influence, and explain observed propagation speeds. These results further support our observations in characterizing the IMW process as self-sustained through the existence of an IMW-calving feedback. This study therefore highlights the importance of short-term ice mélange dynamics in the longer-term evolution of Greenland outlet glaciers.

## 1 Introduction

Greenland's ice discharge is currently contributing approximately half to the total mass loss of the ice sheet (Shepherd et al., 2020), alongside with surface melt. While complex, the interactions between tidewater glaciers and the ocean have been identified as a set of key and important processes involved in Greenland's current and future ice loss.



The high surface speed of Greenland's major outlet glaciers coupled with sustained frontal ablation is resulting in the release of large quantities of ice in proglacial fjords. These fjords have been shaped over thousands of years by fast and concentrated ice flow and are often consisting of deep and narrow corridors. The combination between a narrow outlet and high ice discharge can therefore result in the congestion of icebergs in the fjords of tidewater glaciers. This dense mixture of icebergs and sea ice is commonly and hereafter called ice mélange (sikussak in Greenlandic).

Dense ice mélange often covers the fjords at the front of Greenland outlet glaciers, behaving like a very coarse granular material (Cassotto et al., 2021; Burton et al., 2018). Such ice covers have been identified to have high enough yield strengths to exert back stress on glacier termini (Amundson et al., 2010a; Cassotto et al., 2015, e.g.,). Walter et al. (2012) estimated the back stress from winter sea-ice mélange to range from ∼30 to 60 kPa on the entire face of the terminus of Store Glacier, corresponding to a ∼240–480 kPa mélange-glacier contact pressure Todd and Christoffersen (2014). However, such observation-based

inference of mélange back stress remain scarce and results in low constraints for numerical models. Through this process known as buttressing, ice mélange can strongly influence glacier dynamics (Dupont and Alley, 2005; Amundson et al., 2010b; Robel, 2017; Howat et al., 2010; Nick et al., 2010; Cook et al., 2014; Krug et al., 2015, e.g.,). At a seasonal scale, ice mélange has been shown to promote glacier advance in winter and to further leave the calving front unprotected as it breaks up in spring, leading to higher calving rates and therefore accelerated glacier retreat (Todd and Christoffersen, 2014; Moon et al.,

2015; Kehrl et al., 2017; Joughin et al., 2008; Amundson and Burton, 2018).

Bevan et al. (2019) studied the variations in the position of Kangerdlugssuaq glacier terminus in relation to fjord dynamics using satellite and reanalysis data with a focus on the period from 2011 to 2019. The authors showed interannual warming shelf waters highly reduced the overall strength of the ice mélange making it less efficient in inhibiting calving. In the discussion, they further identified the propagation in the up-fjord direction of one weakening wave in the ice mélange during an entire month in

February-March 2018 using a series of Sentinel-1 radar images. This wave consisted in the progressive retreat of the boundary between a dense jam-packed ice mélange strongly coupled to the calving front and a weaker, sometimes discontinuous, ice cover extending further downfjord. Once the jam-packed ice cover reached a critical – unknown – mass and extent at the glacier terminus, large calving events were observed. While the authors noted this pattern is commonly seen in summer and has recently also occurred in winter, this single example only served as supporting observation for the study of interannual

variations in ice mélange conditions, without further analysis.

Xie et al. (2019) studied such an ice mélange weakening (hereafter IMW) episode that occurred in June 2016 at the front of Jakobshavn Isbræ using a terrestrial radar interferometer. The authors focused on the retrieval of elevation changes through time and showed the IMW front was characterized by an abrupt surface step change of ∼10 m between the thick jam-packed ice mélange and the thinner and weaker ice cover extending in the down-fjord direction. This important surface drop further

explains the strong contrast visible in satellite imagery across the discontinuity, e.g. in the case of the event described in Bevan et al. (2019) at Kangerdlugssuaq glacier. (Xie et al., 2019) also observed an up-fjord migration of this boundary through the occurrence of several collapse-like events, as well as the initiation of large-scale calving once the ice mélange mass reached a critical minimum thickness and extent. This study therefore brought a high resolution characterization of one IMW episode, complementing scarce previous work using satellite imagery.



Our knowledge of IMW episodes in Greenland and their relation to calving activity is therefore based on a highly limited number of studies. The latter mainly investigated multiannual to seasonal patterns using spaceborne observations, or single events using high spatial and temporal field measurements over short periods. A detailed assessment and characterization of short-lived IMW episodes is therefore currently missing.

    In this study, we use successive Sentinel-1A and B images, making the most out of its daily to two-day revisit time over
Greenland, to analyze a sample of IMW episodes at the front of three main Greenland outlet glaciers: Kangerdlugssuaq Glacier, Helheim Glacier and Sermeq Kujalleq in Kangia (Jakobshavn Isbræ). For Kangerdlugssuaq Glacier which features high ice mélange dynamics, we further extend our analysis to include a continuous monitoring of IMW episodes at this location during the June-to-September period from 2018 to 2021. We finally explore the drivers and controls of the IMW process using a simple numerical model for ice mélange motion. We therefore aim at improving the characterization of these short-lived events
to ultimately better understand their impact on the longer-term glacier terminus stability.

## 2   Study sites

Kangerdlugssuaq (also known as Kangerlussuaq) Glacier (hereafter KG; 68.5°N, 33.0°W) is a major outlet glacier situated in the South Eastern sector of the Greenland ice sheet (Figure 1a and c). After minor thinning from 1981 to 1998, KG thinned by ∼100 m from 2003 to 2005 (Khan et al., 2014) and suddenly retreated by ∼6 km in 2005 doubling its surface speed x(Luckman
et al., 2006). In 2011, KG slowed down and started to experience large seasonal variations of more than 3 km in its terminus position and slightly advanced by ∼200 m until 2016 (Kehrl et al., 2017). In 2016, an almost continuous retreat was initiated. KG failed to advance in winters 2016/17 and 2017/18 (Bevan et al., 2019) and reached a terminus position unprecedented in observation records at this time (i.e. since 1932, Brough et al. (2019)). Over the past couple of years, KG seems to have returned to its pre-2016 ice discharge regime (Mankoff et al., 2020) but is still experiencing a retreat of its summer minimum
front position. It currently features a ∼5 km wide calving front which has been suggested to be close to flotation (Bevan et al., 2019). KG calves into a ∼18 km long ∼5 km wide secondary fjord artery, part of a 75 km long 5–10 km wide main fjord. (Murray et al., 2010; Sutherland et al., 2014).

    Helheim Glacier (hereafter HG; 66.4°N, 38°W), ∼400 km South from KG glacier along the South-Eastern Greenland coast (Figure 1a and d), also underwent a major retreat at the beginning of the century essentially between 2003 and 2005 (Luckman
et al., 2006; Stearns and Hamilton, 2007; Howat et al., 2005, 2008). Similarly to KG, it further stabilized from 2006 until a significant increase in solid ice discharge was initiated in 2016 (Mankoff et al. (2020); see Figure 1e). This regime of high discharge lead HG to nearly overtake JI as Greenland's main solid ice contributor to sea level rise in early 2021 and currently appears to be maintained although showing a potential decreasing trend (Mankoff et al. (2020); see Figure 1e). HG is discharging ice into a side fjord (∼20 km long, ∼5.5 km wide) of the main Sermilik fjord (∼80 km long and ∼6 km wide;
Sutherland et al. (2014)). Recent research suggests the front of HG is currently close to flotation and about to initiate a period of rapid retreat Williams et al. (2021).



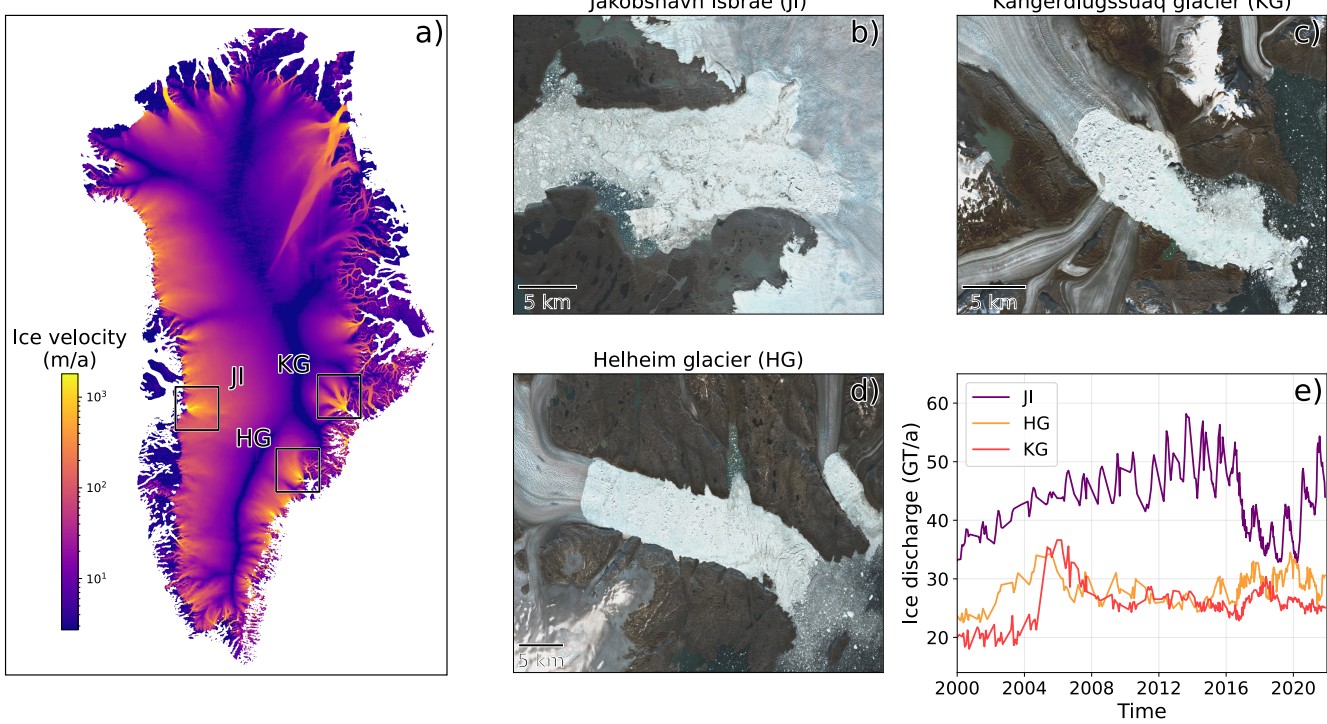

**Figure 1.** (a) Locations of the glaciers of interest shown on a 2020 surface velocity map of the Greenland ice sheet (Joughin, 2021). (b, c, d) Sentinel-2 enhanced true color images (Copernicus Sentinel data 2022, processed by ESA) of their respective calving fronts as least-cloudy mosaics in August 2020. (e) 2000-2020 respective solid ice discharges (Mankoff et al., 2020).

Jakobshavn Isbræ (hereafter JI; 69.2°N, 49.6°W, Fig. 1), situated in Center West Greenland (Figure 1a and b), is among Greenland's fastest glaciers (Joughin et al., 2014). It features a ∼25 km long calving front currently discharging more than 50 Gt of ice per year into the ocean (Mankoff et al., 2020) in the form of large icebergs which then travel through a ∼60 km long ∼5-12 km wide fjord before reaching Disko Bay. JI is directly flowing into the main fjord artery to the open ocean unlike KG and HG which first flow into secondary fjords. JI has maintained a high flow speed of $7\,\text{km}\,\text{a}^{-1}$ since 1875 (Weidick and Bennike, 2007) before the step-wise disintegration of its floating terminus after 1997 (Sohn et al., 1998) resulted in a twofold increase of the surface velocity in the main trunk (Rignot and Kanagaratnam, 2006). After a recent slowdown linked to colder ocean waters between 2016 and 2018 (Khazendar et al., 2019), JI has now returned to its regime of sustained mass loss since 2020.

JI, HG and KG are currently the top three contributors to Greenland's solid ice discharge, accounting for 22% of the total discharge in late September 2021 (10, 6 and 5%, respectively; Mankoff et al. (2020)) and are associated with a potential sea level rise of ∼1.3 m (Kjeldsen et al., 2015). Improving our understanding of the dynamics of those major outlet glaciers is therefore crucial to better resolve the current and future mass loss of the Greenland ice sheet.





## 3 Data and Methods

### 3.1 Satellite imagery

We used Sentinel-1A and B synthetic-aperture radar (SAR) acquisitions of backscatter signal amplitude in polarized (Horizontal-Vertical; HV) and non-polarized (Horizontal-Horizontal; HH) modes during the June-November period from 2018 to 2021 to monitor ice mélange conditions and dynamics at the front of the three glaciers of interest. Associated to the high latitude of the study areas, a revisit frequency from one to four days (median frequency of one day) was achieved, without shading from clouds to which radar satellites are immune. The combination of polarized and non-polarized modes allowed for the detection of variations in ice mélange characteristics, which could remain undetected using one mode alone. The high spatial and temporal heterogeneity of ice mélange, as well as the potential variety of processes affecting it, make the HV mode particularly useful for surface characterization. In combination with Sentinel-1 radar observations, we used Sentinel-2 optical images acquired with a frequency of one to three days over Greenland.

All satellites images used in this study were downloaded with the earthspy Python package (Wehrlé, 2022), a wrapper download tool for Sentinel Hub services (Sentinel Hub, 2022).

### 3.2 Calving event detection

We manually detected the timing of large-scale calving events at the front of the three glaciers of interest using a combination of Sentinel-1 radar and Sentinel-2 optical data to increase temporal coverage and resolve potential detection ambiguities. We further qualitatively quantified the magnitude of each calving event in a simplified manner, depending on the detached area visible in spaceborne observations. This simple proxi for calving event magnitude could take values of 1, 2 and 3 for small-, medium- and large-size calving events, respectively.

### 3.3 Tracking of IMW propagation

The clear signature of IMW episodes in Sentinel-1 data at the front of KG (see supplementary videos S1 to S4, left panels) allowed for a mapping of the discontinuity between jam-packed ice mélange in contact with the glacier terminus and weaker ice mélange extending further down-fjord. HG also features frequent IMW episodes every year, but their propagation is often less clearly visible, and can remain ambiguous. We therefore decided to restrict the analysis at the front of HG to three well discernible IMW episodes. A similar strategy has been followed for JI.

The results consist of a catalog of line objects representing the positions of the IMW fronts through time, and associated meta data (date, terminus position) for each of the three glaciers. The line objects are stored in shape files in the NSIDC Sea Ice Polar Stereographic North projection (EPSG:3413).

Manual detection was chosen after limited efforts to develop an automated detection of IMW episodes. Trials were focused on an unsupervised area classification through K-Means clustering using spatial coordinates as well as Sentinel-1 backscatter amplitude in HH and HV modes as features. Performances were not satisfactory enough for the method to be used in this study,





especially in complex situations which often could not be resolved with this type of surface characterization alone. Taking the dynamics of the ice mélange cover into account by adding e.g. surface velocities as another feature to distinguish between ice surface types would likely give more satisfactory results but was not further investigated here.

### 3.4 Biased Random-Walk Model

To better understand the dynamics of dense ice mélange, we developed a simple numerical model based on the idea that the floating ice blocks move like in a one-dimensional Brownian random walk. This model, called BRIMM (biased random-walk ice mélange model) is based on discrete blocks, representing floating icebergs, that move along the axis of the fjord. Blocks are created at the glacier terminus by calving, and float away when they reach the end of the fjord. At each time step, each block moves by a random distance in a random direction (up- or down-fjord). We ignore cohesion and momentum transfer between

blocks. This means that if one block moves into another, it is simply stopped in contact with that other block, but does not affect the position or motion of the other block.

At each model time step, all blocks move according to a random walk with uniformly distributed distance and random direction of motion (up- or down-fjord). To achieve an overall motion down-fjord, a small bias is added to the random distribution, making the blocks more likely to move away from the glacier terminus. The leftmost block, representing the advancing glacier

terminus, moves at a constant speed.

The model time step size was chosen as 1/50 of a day (about 0.5 h), which is in the order of the time scale needed for acceleration and subsequent stopping of a large iceberg. The maximum random motion $\Delta x_{\mathrm{max}}$ of the blocks at each time step was varied between 20 and 160 m. The distance of random motion $\Delta x_r$ was calculated by taking a value $p_r$ from a uniform random distribution, and by altering it by a bias $p_b$

$$\Delta x_r = \Delta x_{\mathrm{max}} \, 2 \left( p_r - 0.5 + p_b \right) . \tag{1}$$

To obtain a net motion away from the calving front, a bias $p_b$ between 0.01 .. 0.11 was added to the values from the uniform random distribution $p_r$ (between 0 and 1). ($p_r \in [0..1]$)

Two calving criteria were implement to reproduce in a simple manner the fact that dense ice mélange prevents the glacier terminus from calving if the mélange is closely packed. First, calving happens when the block closest to the calving front

moves a certain distance away from the front. This emulates open water between ice blocks in proximity of the terminus, and corresponds to a unconsolidated mélange without any stress transfer.

The second condition is fulfilled when within the last 5 km in front of the terminus a lead opens between two blocks that is wider than 1.5 block widths. This can be thought of as emulating the collapse of an arch structure buttressing the terminus. The length scale (5 km) corresponds to the width of the fjord since such an arch is usually shaped elliptically.

During each calving event the glacier terminus moves back by a certain distance. A prescribed number of new ice blocks (10 to 30 in our model runs) that are initially vertical with an along-flow length of 20 m is created. During calving, the blocks turn around and extend 100 m horizontally, therefore taking up more space, and pushing away blocks from the calving front that would otherwise overlap. The net effect is a dense ice mélange that extends in front of the terminus (Robel, 2017).





The BRIMM model is characterized by a set of parameters which values are chosen to represent the processes in a fjord. All
parameters and their limits used in this study are shown in Table 1.

| Parameter | symbol | std. value | min | max |
|---|---|---|---|---|
| length of fjord | | 50 km | | |
| length of floating block | | 100 m | | |
| length block before calving | | 20 m | | |
| flow speed of the glacier terminus | | 15 m/d | | |
| number of tracked floating blocks | | 300 | | |
| number of calving blocks | | 20 | 10 | 30 |
| length of terminus retreat after calving | | 20*20 = 400 m | 10*20 m | 30*20 m |
| random walk time step | $\Delta t$ | 0.02 d | | |
| maximum random walk step length | $\Delta x_{\max}$ | 50 m | 10 | 80 |
| random walk bias | $p_b$ | 0.02 | 0.01 | 0.11 |
| calving criterion 1: width of lead | | 20 m | | |
| calving criterion 2: width of lead | | 150 m | | |

**Table 1.** BRIMM model parameters and their ranges explored in this study.

The BRIMM model was implemented in the Python programming language and is publicly available under the GPL-v3
license (`https://github.com/MartinLuethi/BRIMM/`).

## 4   Results

### 4.1   IMW episodes at Kangerdlugssuaq Glacier

Figure 2 shows three IMW episodes through jam-packed ice mélange at the front of KG. The upper panels delineate positions
of the transitions between jam-packed and weak ice mélange or open ocean, color-coded for different points in time. The
background Sentinel-1 HH images was acquired in the middle of each IMW episode. The lower panels show the mean along-
fjord distance of the IMW fronts through time. Distances were computed from a constant on-ice reference point situated 1 km
upstream of the most retreated glacier terminus position mapped in this data set.

In all three examples, the IMW front gradually propagates up-fjord in direction of the glacier terminus after a first detection
close to the secondary fjord's mouth (∼17 to 20 km from reference). Interestingly, the three episodes are associated with
relatively variable propagation durations (38, 25 and 32 days for the 2018, 2019 and 2021 episodes, respectively) and patterns.
In the case of the 2018 and 2019 episodes, the IMW front consisted of the interface between a dense, jam-packed ice mélange
(up-fjord) and a weakened ice mélange cover (down-fjord) which gradually disintegrated through collapse-like events. This is
particularly visible on the Sentinel-1 image in Figure 2a where the jam-packed mélange in contact with the glacier terminus
appears relatively dark, while the down-fjord area, consisting in a weak and loose ice mélange, appears lighter before the




open water featuring an almost vanishing backscatter intensity (specular reflection) prevails. In contrast, during much of the 2021 episode the IMW front consisted in a clear-cut discontinuity between jam-packed ice mélange and open water, without a weakened ice mélange cover in between the two types of surfaces.

The 2019 and 2021 episodes feature a clear kink in the propagation curve (on 2019-07-16 and 2021-09-25, respectively; Panels 2e,f) corresponding to a transition from fast to slower propagation around 10 km from reference. The 2018 episode (Panel 2d) shows a more gradual propagation with a period of small position variations at the fjord mouth (2018-06-23 to 2018-07-02). For all three examples, the fastest propagation between two consecutive IMW front detections occurred when the IMW front was situated in the widest area of the fjord ( $10 - 16$ km from reference) where the ice mélange is the weakest on

average. Similarly, the slowest propagation speeds were observed in close vicinity of the glacier terminus (closer than 10 km from reference) where the ice mélange is thick and dense, and the fjord narrower.

All three episodes were followed by a large-size (2018 and 2019) or medium-size (2021) calving event a few days after the last detection of the IMW front. These observations suggest a causal relation between the disintegration of jam-packed ice mélange and the occurrence of calving events at the front of KG.

**4.2 IMW episodes at Helheim Glacier**

Figure 3 illustrates three IMW episodes in the Sermilik fjord in front of HG. The three episodes were associated with propagation durations that are significantly shorter than at KG (17, 12 and 18 days for 2018, 2019 and 2021 episodes respectively) and all three featured a weak ice mélange cover down-fjord from the IMW front. The 2019 and 2021 episodes show a pattern of fast up-fjord propagation and slower propagation or even stabilization closer to the glacier terminus, similar to the events

presented at KG. The propagation curves feature a kink (on 2019-08-08 and 2021-05-29, respectively), but here at two along-fjord locations ∼3 km apart. The 2018 IMW episode shows a more gradual propagation towards the glacier terminus, similarly to the 2018 episode at KG, with a short stabilization from 2018-05-21 to 2018-05-24.

In contrast to KG's fjord, HG's fjord is of relatively constant width and orientation. This suggests that the strength and density of the ice mélange at HG plays a greater role for its dynamics than the variations in its fjord geometry. The fjord at HG

is, however, connected to a secondary artery on its northern shore which alters the ice mélange structure. This impact is visible through the pinning of the IMW front on this crossing point (Figures 3a-c) at the beginning of the 2019 and 2021 episodes (2019-08-05 and 2021-05-26), and in the middle of the 2018 episode (2018-05-21 to 2018-05-24).

In a manner similar to KG, at HG large calving events followed the last detection of the IMW front. However, at HG also small calving events occurred during the IMW episodes (one in 2019 and 2021; two in 2018). This might suggest that the IMW

process exerts less control on calving activity at the front of HG than KG.

**4.3 IMW episodes at Jakobshavn Isbræ**

Figure 4 shows three IMW episodes at the front of JI These three episodes show a wider variety of propagation characteristics than those at KG and HG. Propagation durations of 10, 27 and 65 days were determined (for the first and second episode of 2018, and the 2021 episode, respectively) therefore showing a large range of 55 days.



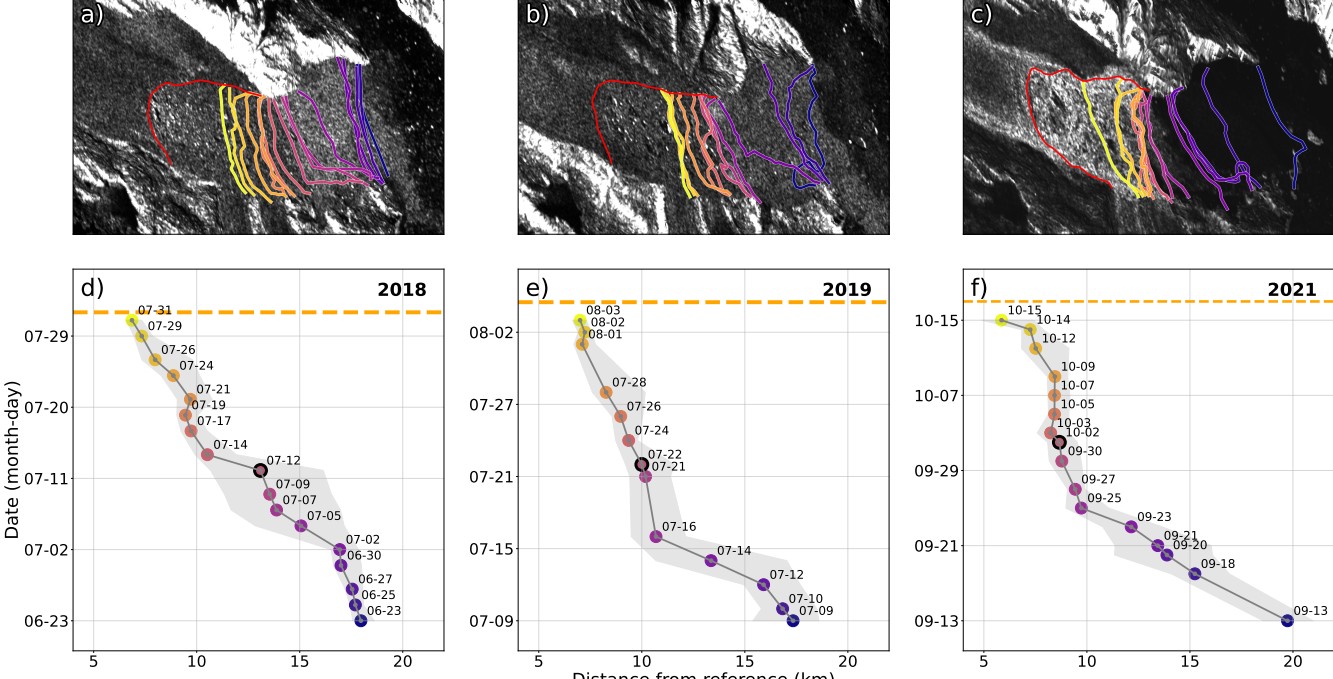

**Figure 2.** Three IMW episodes at KG. (a, b, c) Positions of the IMW fronts through time in cartesian coordinates, overlaying Sentinel-1 HH images acquired half-way through each IMW episode. Red lines correspond to the position of the calving front. (d, e, f) Distance between IMW fronts and an on-ice reference situated 1 km upstream from the most retreated glacier terminus position mapped in the data set. Dot color corresponds to each IMW front mapped in (a, b, c) through time and black dot contours to the acquisition dates of the respective Sentinel-1 HH images. The grey curves indicate the median along-fjord distance of the IMW front from the on-ice reference. The envelopes correspond to the range between minimum and maximum along-fjord distance at each time steps (therefore quantifying the across-fjord variations in IMW front position). Horizontal orange dashed lines indicate the timing of calving events detected with spaceborne observations, their width being proportional to their approximated magnitude (small, medium or large). Time is running from bottom up.

The first 2018 IMW episode featured a very clear up-fjord weakening propagation associated with IMW fronts of low complexity an limited across-fjord kinking. This episode ended only 4 days before the beginning of the second 2018 episode which was almost 3 times shorter (10 days) but also showed a clearly visible propagation. During both episodes, the dense ice mélange covering an embayment on the southern shore of the fjord was not affected. The 2021 episode shows more complex IMW front outlines which may be linked to the breakup of the ice mélange in the embayment area, suddenly increasing the distance between the two lateral pinning points and therefore most likely contributing to a more variable and unstable discontinuity.

For the first 2018 episode and the 2021 episode, a kink in the propagation curve is again visible (on 2018-05-11 and 2021-07-29) but this time with the fastest propagation occurring in the vicinity of the glacier terminus, unlike the events presented at KG and HG. In the case of the second 2018 episode and the 2021 episode, the Sentinel-1 images in the middle of the two



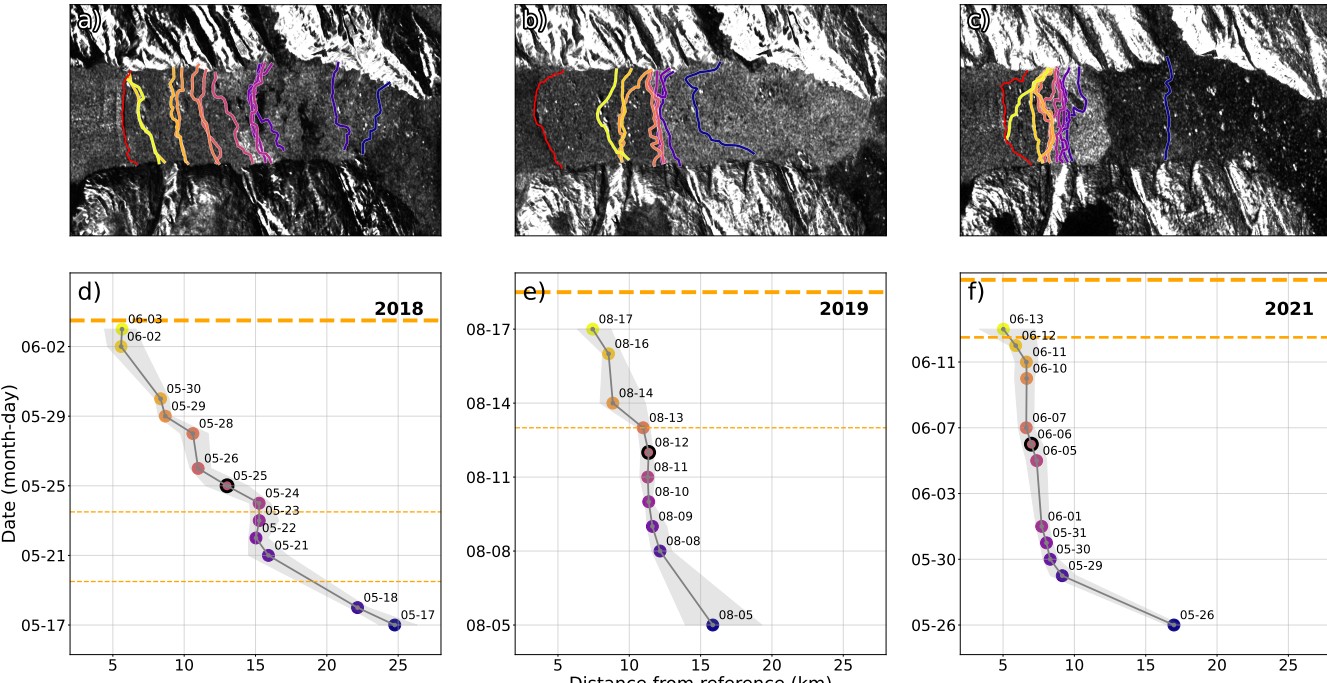

**Figure 3.** Three IMW episodes at HG presented in the same manner as in Figure 2. See caption of Figure 2 for details.

events show the formation of two polynyas (open water, visible as dark areas surrounded by ice mélange) on the southern and northern fjord shores, respectively. The polynyas were formed right in front of the IMW discontinuity highlighting a strong decoupling between areas of dense and weaker ice mélange.

Similarly to KG and HG, large-size (first episode in 2018 and 2021 episode) and medium-size (second episode in 2018) calving events occurred at the end of the respective IMW episodes. In the case of the 2021 episode, a medium-size calving event occurred in the early stage of the weakening propagation.

### 4.4 Continuous IMW episode analysis at Kangerdlugssuaq glacier

To better understand the main characteristics of the IMW episodes in the KG fjord, IMW fronts were tracked continuously during the June-November periods of the years 2018 to 2021. This monitoring shows a highly dynamic pro-glacial ice cover during summer and fall.

Black lines in Figure 5 show the along-fjord propagation of IMW fronts. The IMW episodes displayed in Figure 2 are highlighted with red vertical bars. Orange horizontal lines indicate calving events visible in satellite imagery, as well as an estimate of their magnitude.

During the study period of four summers and falls, a total of 30 IMW episodes were observed. The propagation of the associated IMW fronts is well visible in the animations (supplementary videos S1 to S4 from 2018 to 2021). 5 to 8 IMW



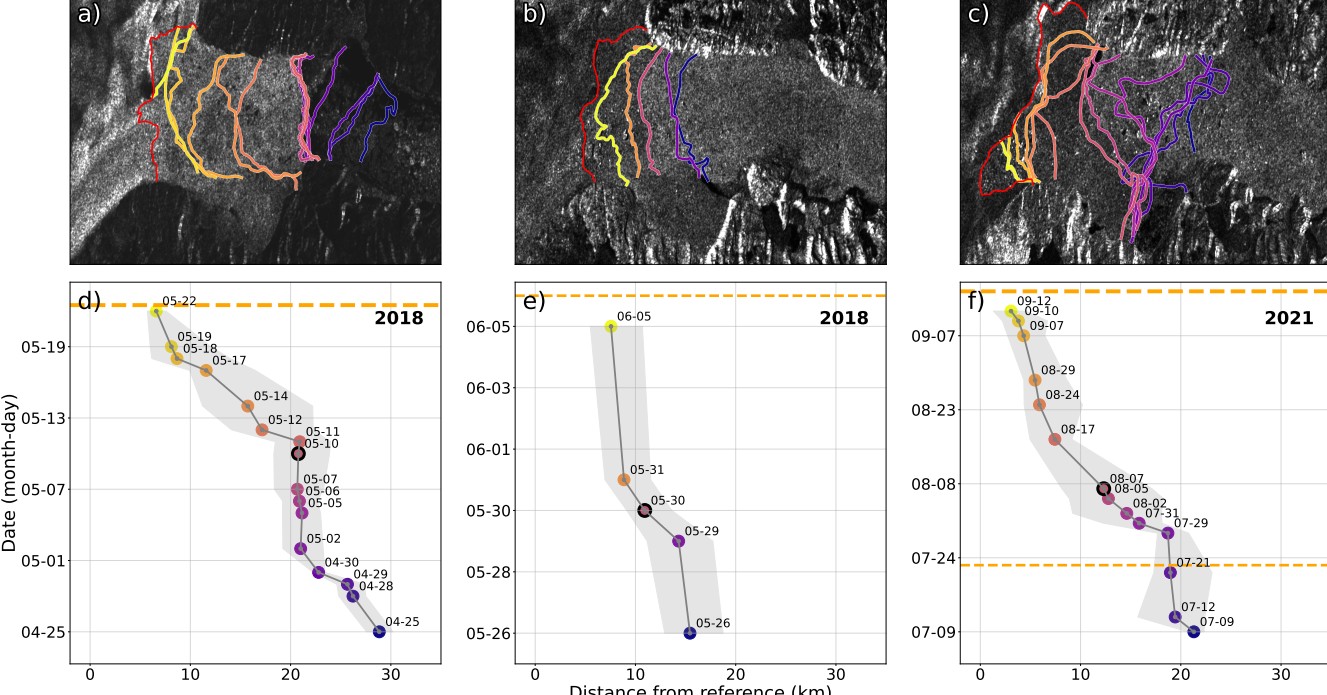

**Figure 4.** Three IMW episodes at JI presented in the same manner as in Figure 2. See caption of Figure 2 for details. The Sentinel-1 HH images have been flipped horizontally to keep the right-to-left convention for IMW propagation direction.

episodes were detected per season (June to November), corresponding to an average recurrence time of $24 \pm 3$ days. During the same time 37 calving events were detected using space-borne observations with an average of $9 \pm 1$ events per season. Combining the IMW tracking with calving event detection, we found that 87% of the IMW episodes were closely followed by a calving event. Conversely, we found that 70% of detected calving events occurred subsequent to an IMW episode. Assuming that IMW episodes can have a cascade effect, and including also calving events occurring a few days after the IMW-triggered calving events, this ratio increases up to 81%. A good example of this cascade effect is late November 2020 (at top of Fig. 5c).

Using our simple proxi for calving magnitude, we found that from 78% to 90% of the cumulated calving magnitude over the study period was released following an IMW episode. While the strong link between calving activity and the termination of IMW episodes seems clear, the calving inhibition during those episodes is even more pronounced. Only 16% of the calving events occurred during an IMW episode before the IMW front reached the glacier terminus, and only 7% of the cumulated calving magnitude was released during IMW episodes.

Focusing on intra-episode characteristics, a recurrent pattern (45% of all IMW episodes) of fast propagation speeds at the early stage of the episodes (down-fjord), and slower propagation speeds at the end of the episode (in the vicinity of the glacier terminus) was observed. Such a pattern was already identified in the 2019 and 2021 episodes presented in Figure 2. Two IMW episodes (June 2019 and October 2019) featured a two-stage propagation without calving. The first IMW front stabilized mid-





fjord at around 9 km from the reference point, and was overtaken a few days later by a second and faster IMW front. No calving event was detected following the intermediate stabilization.

Winter IMW activity was low in three of the four years spanning the study period, with the notable exception of winter 2018/19. In winters 2019/20, 2020/21 and 2021/22 no IMW episodes and only rare calving events were observed. The ice mélange remained tightly coupled to the glacier terminus which was continuously advancing. In winter 2018/19 IMW episodes

occurred without any interruption from the end of continuous monitoring (late November) to its beginning the following year (early June). Also during this period the calving activity showed a similar relation to IMW episodes than during the spring-to-fall period.

These strong interannual variations in winter ice mélange dynamics are likely due to starkly different meteorological conditions. In winters 2019/20, 2020/21 and 2021/21 sea ice remained dense and almost motionless in the fjord for 108-152 days

(∼120 days from 2019-02-02 to 2019-06-02; ∼108 days from 2020-02-15 to 2020-06-02; ∼152 days from 2021-01-18 to 2021-06-19). In contrast no dense sea ice formed at the fjord scale during the entire winter 2018/19, and the ice cover remained relatively mobile.

## 4.5  IMW characteristics

Figure 6 presents several quantities characterizing IMW episodes: propagation distance, duration and propagation speed. Quan-

tities for the three IMW episodes studied at KG, HG and JI (Figs. 2- 4) are shown with colored dots. The distributions obtained from 30 IMW episodes analyzed at KG (Fig. 5) are shown as violin plots (gray areas). The second IMW episode detected in 2021 (Fig. 5d) is not shown in Figure 6c due to its anomalously high propagation speed (3.4 km in one day) which would have highly altered and flattened the visualization of the probability density function. This episode nevertheless still contributes to the median propagation speed.

The median propagation distance determined at KG was 5.9 km for a median propagation duration of 17 days, with a wide spread of values from 1 to 52 days. The median propagation speed was 400 m/d with a significant variability between 100 and 1200 m/d. The three IMW episodes of Figure 2 illustrate the high end of propagation distances at KG (10.3 to 13.9 km), and show propagation duration values that are also above the median but lower than the maximum (25 to 38 days, for a maximum of 52 days). The resulting propagation speeds remained close to the median (variations from +30 to -100 m/d).

While the three isolated IMW episodes studied at HG and JI cannot give any clear insight into the distribution of the propagation characteristics at the front of these glaciers, they still illustrate possible situations that can be compared to the extended detection at KG.

The combined 6 IMW episodes at JI and HG featured higher propagation distances than the median distance computed at KG (from 7.7 to 22.1 km) with three events (two at JI, one at HG) above KG's maximum of 13.9 km. The propagation durations

of the three IMW episodes at HG are relatively close to KG's median duration (from 12 to 18 days) while the IMW episodes at JI show a wider spread (from 8 to 65 days). Five out of six IMW episodes show higher propagation speeds than KG's median, and one episode at JI (21-07-09) was associated with a lower speed of 300 m/d. Similarly to the maximum propagation distance and duration (22.1 km and 65 days), the overall maximum propagation speed (1.5 km/d) was recorded for an IMW episode at JI.





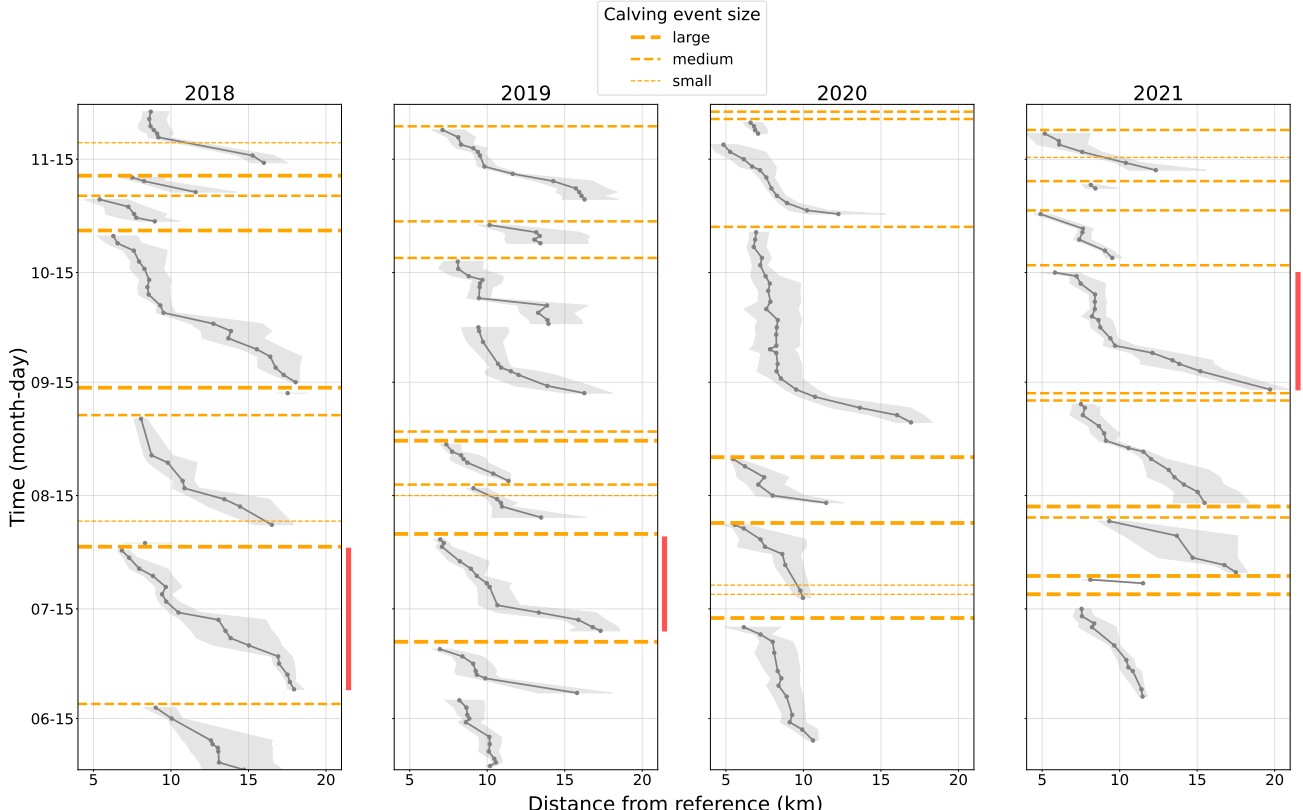

**Figure 5.** Detection and tracking of IMW episodes at KG during the June-November period from 2018 to 2021 presented in a similar manner as in pannels d, e and f from Figure 2 to 4. The three vertical red bars indicate the three episodes presented in Figure 2.

The geometry of JI's fjord (significantly longer and wider than KG's and HG's fjords) can explain higher propagation distances and durations at JI. The conditions to obtain higher propagation speeds might be linked to a lower ice mélange cohesion at JI than KG and HG, but remain mostly unclear at this point.

### 4.6 BRIMM model results

The BRIMM model was run with a set of geometrical and model parameters that are inspired by the physical characteristics observed at KG and JI (Fig. 6), and which were varied in ranges corresponding to realistic values (Tab. 1). The model, based on a random walk of discrete blocks, shows an emergent dynamics that resembles observations (see supplementary videos S5 to S7). The results show iceberg jamming in the fjord after calving, IMW episodes with realistic propagation speeds of the weakening front, and a quasi-periodic, punctuated dynamics. By variation of two model parameters different dynamical characteristics emerge that allow us to better understand the processes controlling ice-choked fjords.





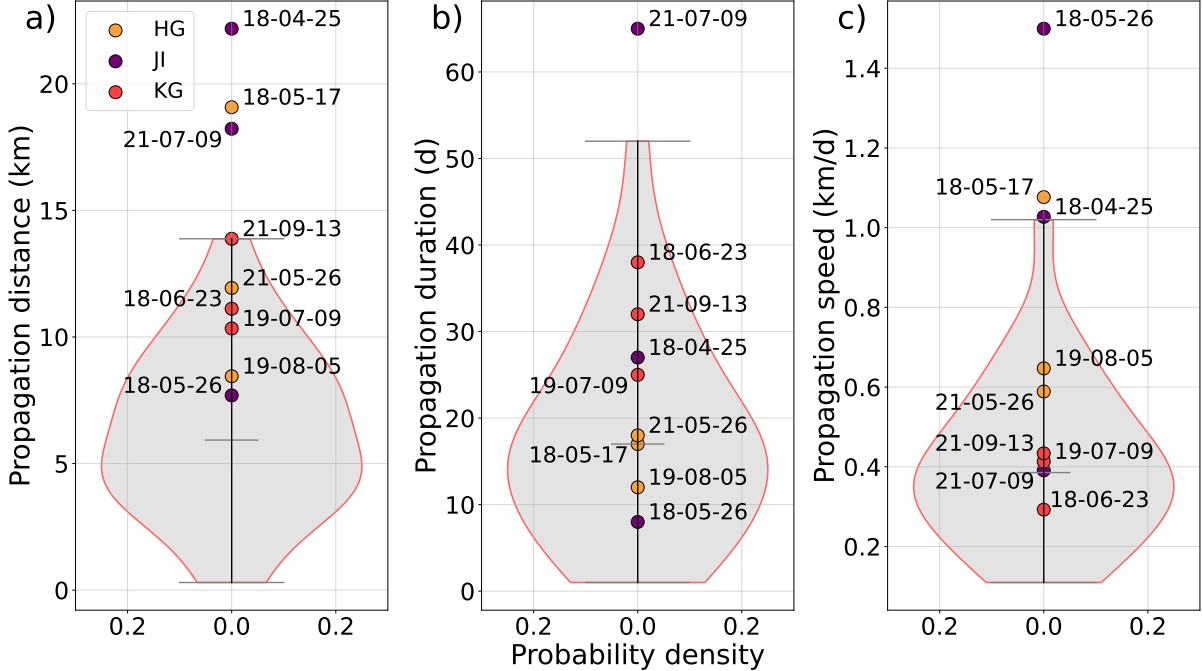

**Figure 6.** (a) Propagation distance, (b) duration and (c) speed of 29 IMW episodes detected at KG, presented as mirrored probability density functions, also known as violin plots (gray areas). The central tick corresponds to the median, and the upper and lower ones to the distribution's extrema. The second IMW episode detected in 2021 (see Figure 5) has been removed from (c) due to its anomalously high propagation speed (3.4 km in one day) which would have highly altered the representation of the probability density function. This episode nevertheless still contributes to the variable's statistics. The characteristics of each set of three IMW episodes analyzed at KG, HG and JI are shown as red, orange and purple dots, respectively.

For all model runs, a fixed number of 300 blocks, representing floating icebergs, was used, although not all blocks are always

within the fjord of 50 km length. The number of blocks released from the glacier at each calving event was set to 10 (or 20, 30). This corresponds to a glacier retreat of 200 m (or 400, 600 m) at each calving event, assuming that blocks are 20 m long before calving. We further assume that all blocks rotate during calving and occupy an along-flow length of 100 m, therefore adding 1000 m (or 2000, 3000 m) of floating icebergs to the fjord in front of the terminus.

From the model experiments we found that the parameters $\Delta x_{\mathrm{max}}$ and $p_b$ (Eq. (1)) are the most important controls of the

emergent dynamics. The maximum random motion at each time step $\Delta x_{\mathrm{max}}$ quantifies the agility of the ice mélange. The random bias $p_b$ controls by how much the random motion is directed out of the fjord and away from the glacier terminus. In what follows, all model results are shown for variations of these two parameters.

Figures 7 and 8 show the BRIMM model results in a manner similar to Figure 5. Clearly, the frequency of calving events depends crucially on both $\Delta x_{\mathrm{max}}$ and $p_b$. With increasing random motion per time step the frequency of calving events

increases. Higher mobility of the blocks decreases the density of the ice mélange and therefore leads to more open water close





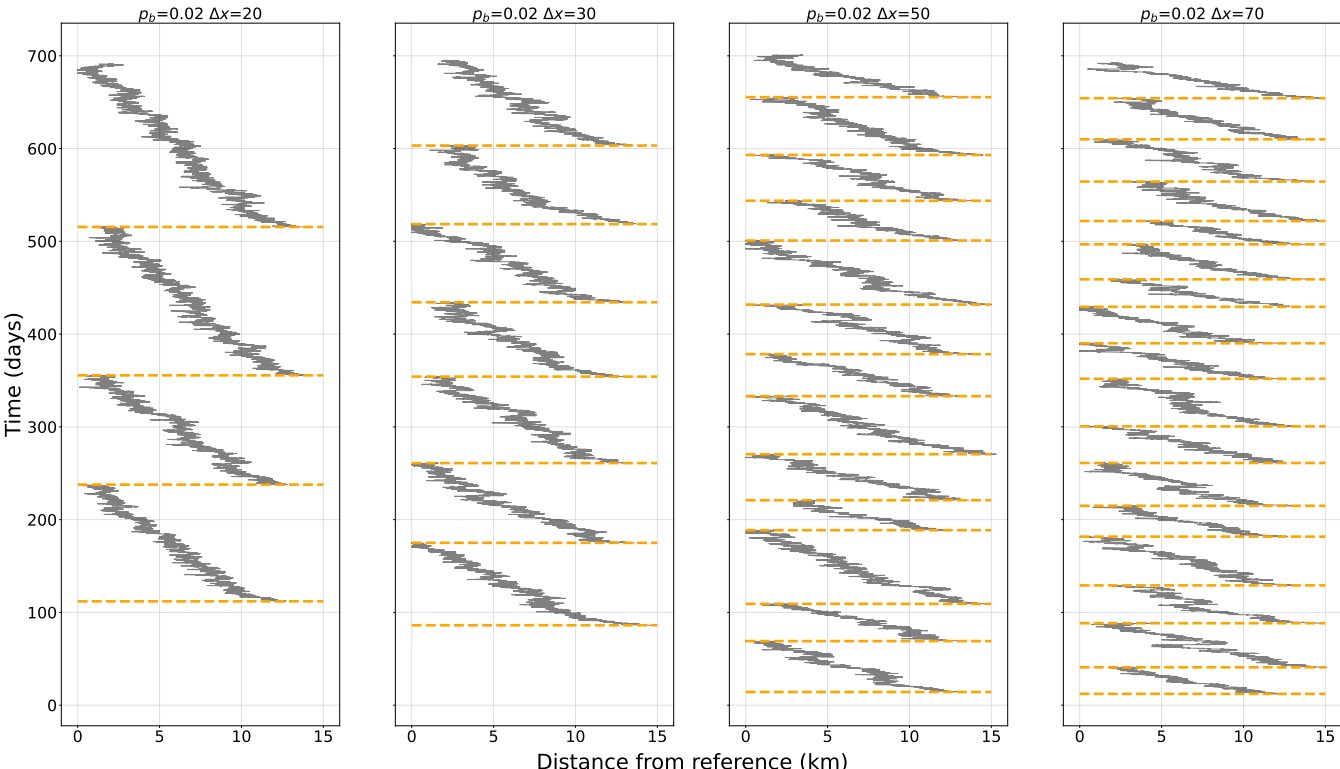

**Figure 7.** Along-fjord propagation of IMW fronts as modeled by BRIMM. In each panel a different value of maximum random motion $\Delta x_{\max}$ was used.

to the terminus, triggering calving events according to our model assumptions. Similarly, a higher random bias moves the icebergs at a faster speed away from the glacier terminus, again leading to more open water and more frequent calving.

Several quantities that can be compared to observations were extracted from the BRIMM model results: the average advance rate of the glacier terminus position, the propagation speed of IMW fronts, the frequency of calving events, and the mean duration of the IMW episodes. Comparing these quantities to observations allows us to determine model parameters that reproduce realistic dynamics. Figure 9 shows these characteristic quantities color-coded for variations of the model parameters $\Delta x_{\max}$ and $p_b$. Red lines indicate the observed ranges from Figure 6.

The most important prerequisite for a calving glacier in a fjord is a relatively stable terminus position (otherwise there would be a glacier extending to the fjord mouth, or no glacier at all). Figure 9a shows that for achieving such a stable terminus either the random motion or the bias has to be large. This shows that the action of these parameters is complementary. Similar conclusions can be drawn from the IMW propagation speed (Fig. 9b) and the average duration of a IMW episode (Fig. 9c).



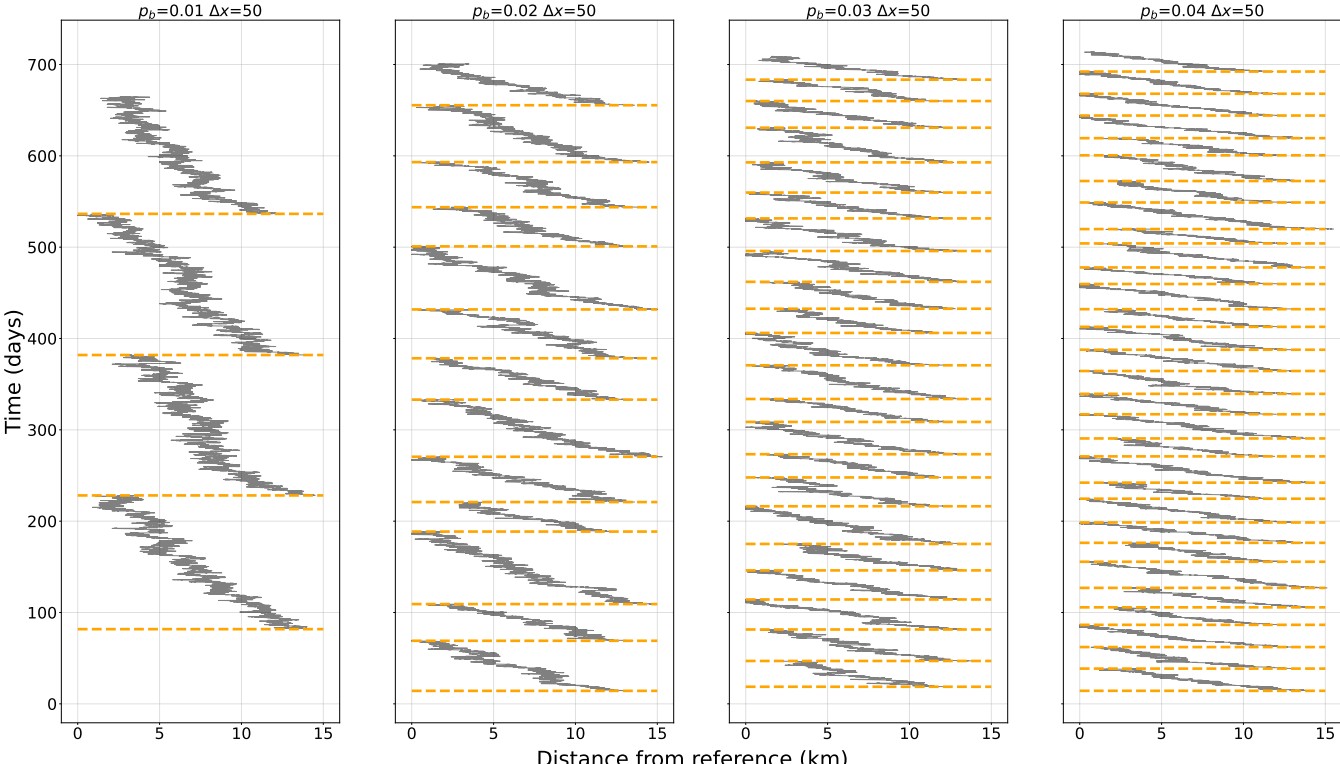

**Figure 8.** Along-fjord propagation of IMW fronts as modeled by BRIMM. In each panel a different value of the random bias $p_b$ was used.

## 5 Discussion

Through the inspection of Sentinel-1 and Sentinel-2 spaceborne observations at the terminus of JI, HG, and KG, we analyzed a set of IMW episodes and linked them to the timing of large-scale calving events. Our results conclusively show that dense
jam-packed ice mélange is an efficient short-term calving inhibitor. Removing or weakening this dense ice mélange in front of the glacier terminus by a propagating IMW front releases the inhibitor, and therefore effectively triggers calving. In this sense, the IMW process can be understood as an important control on calving activity.

### 5.1 Self-sustained IMW cycle

While the role played by dense ice mélange in inhibiting calving at the seasonal scale has been discussed before (e.g. Walter
et al., 2012; Cassotto et al., 2015; Bevan et al., 2019; Cook et al., 2014) and still receives a lot of attention, the general absence of calving events during the cyclic propagation of successive IMW fronts over the summer period appears as a more complex relation and raises important questions.





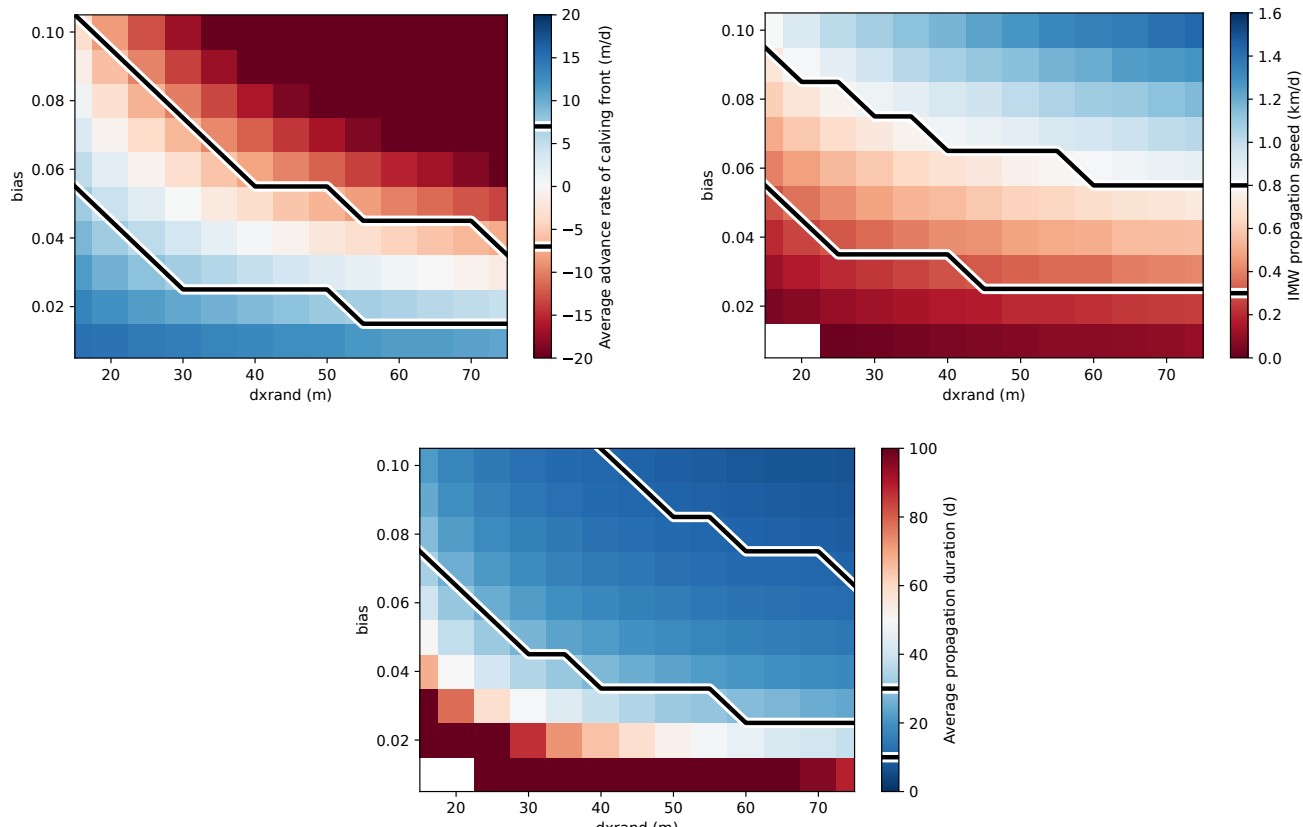

**Figure 9.** Dependence of observable quantities in the BRIMM model on model parameters $\Delta x_{\max}$ and $p_b$. Black lines indicate the range of observed values from Figure 6.

How can the ice mélange initially inhibit and its disintegration further trigger calving in a cyclic manner throughout the season, and why is its influence not suppressed after the first IMW episode? In other words, how can a weakened ice mélange
switch back to its strong inhibiting behavior?

Based on our observations and the results from the BRIMM model, we suggest that the jam-packed ice mélange initially plays a similar role at short time scales than observed in seasonal patterns, until it reaches a minimum length close to the glacier terminus. At this stage, the suppression of calving by the ice mélange is no longer strong enough to prevent calving events that would have been ready to occur without such a support. The subsequent calving event leads to large changes of the stresses
within the glacier terminus, potentially leading to a cascade effect such that secondary calving events may occur eventually (Fig. 5).

The switch back to a strong inhibiting behavior of the ice mélange and the resetting of its distribution within the fjord are needed for the next IMW episode to occur. We suggest that the resetting is initiated by the calving event itself. Each calving event releases large amounts of floating ice into the fjord, thus strengthening the ice mélange in front of the terminus





(Xie et al., 2019). Relaxation within this dense ice forces the expansion of a spatially constrained ice cover, and therefore increases its density and cohesion. Buoyancy-driven calving events, as well as chunks of ice with a low length-to-height ratio before detachment, have the strongest jamming effect as their capsizing strongly increases their area at the fjord surface. Such capsizing icebergs are frequently observed at the fronts of major Greenland outlet glaciers which are close to flotation and reside in very deep fjords. In addition, the continuous advance of the glacier terminus also promotes the densification of the

proglacial ice mélange.

Even more than increasing the density of the ice mélange, its cohesion can be overcome in such conditions and force its yielding starting at the down-fjord boundary: a new IMW episode is therefore initiated. Using a discrete element model to simulate mélange as a cohesive granular material, Robel (2017) showed that the occurrence of calving events initiates a propagating jamming wave within the mélange (in the down-fjord direction, opposite to IMW episodes) causing a local compression

and slow mélange expansion. The authors further describe this jamming wave as the trigger of widespread fractures in the sea ice. This result further supports our hypothesis that calving is resetting the dense ice mélange, thus sustaining the IMW process.

We therefore suggest that the intraseasonal IMW process remains self-sustained following the spring onset and until winter conditions prevail. The latter condition might never be attained as exemplified in the warmer-than-usual winter 2018 when the IMW process never shut down at KG. This cyclic behavior therefore consists in a IMW – calving feedback where IMW

episodes control the timing of calving events, and calving events reset the fjord to conditions triggering the next IMW episodes.

**5.2 Prerequisites for the IMW process**

In order to better understand whether IMW episodes play a role at other outlet glaciers, we qualitatively inspected ice mélange dynamics in a large selection of Greenland outlet glaciers.

The main initial condition for IMW episodes is the sustained presence of a dense ice mélange. Such dense, jammed-packed

ice mélange could only be observed at the front of fast outlet glaciers discharging into relatively narrow fjords. We therefore suggest the formation of such a dense ice mélange is best promoted when high ice discharge into the fjord is combined with a narrow outlet, limiting the evacuation of floating ice to the open ocean. Many Greenland outlet glaciers, while associated with a high ice discharge, mostly fail at retaining icebergs all year long. These glaciers, such as Store Gletscher, are flowing into a fjord that is not narrow enough with respect to the incoming ice flux to create a dense ice cover. We also suggest that

there is no absolute value of fjord geometry and solid ice discharge for the formation of dense ice mélange but rather different combinations of parameters. This implies that a ratio between solid ice discharge and fjord's narrowness might give valuable insight into ice mélange conditions at a given location. For a more realistic representation, the dependence of ice mélange density on air and ocean temperature should be included. It is noteworthy that hardly any dense ice mélange was observed in South Greenland where the temperatures are the highest on average. Combining these three primary parameters, the three

glaciers of interest in this study appear as clear candidates for the presence of a dense ice mélange cover, which is supported by in-situ and remote sensing observations. More specifically, KG emerges as the best candidate as it features the highest ratio of discharge to fjord width. KG is also the glacier in our selection featuring the most frequent and clearest IMW episodes. This suggests that the higher the ice jamming is, the clearer the IMW patterns appear. While we presented here what we





consider as the primary conditions for the formation of dense ice mélange, we acknowledge that the latter also depends on
ocean temperatures as well as on iceberg size distributions, among many other factors.

A search for IMW episodes in a large selection of outlet glaciers around Greenland yields interesting results. Clear IMW episodes were found at Alison glacier, Upernavik Isstrøm, Sverdrup glacier, Fenris glacier, Nansen Glacier, Anoritup Kangerlua and in Mogens Heinesen Fjord. While the IMW process is clearly observable, the IMW activity at these glaciers is lower than at KG, with a number of events comparable to HH and JI. We suggest that this observation is mainly linked to the conditions
for dense ice mélange discussed above, which are only partially fulfilled at those locations.

With a better insight into the conditions for a sustained dense ice mélange and a wider view of IMW dynamics around Greenland, it is now important to discuss the actual drivers of IMW episodes. Bevan et al. (2019) showed that the recent interannual ice mélange dynamics at KG were strongly impacted by the warming of shelf waters. Here, we used the ERA5 global reanalysis (Hersbach et al., 2020) in search for environmental parameters influencing the shorter-term ice mélange
dynamics at the IMW episode scale. We analyzed air temperature, wind direction and speed as well as sea surface temperature variations, wave heights and tides at the mouth of KG's fjord and yet could not find patterns matching the timing, duration (17 days on average) or frequency (average recurrence time of 24 days) of the IMW episodes presented in this study, and this with or without delays. Further comparing the conditions at the onset of IMW episodes with the average seasonal conditions during the study period using a Student's t-Test, we did not find any statistically significant differences. This absence of relation might
be linked to ERA5's incapacity to resolve the smaller scale dynamics along the coast of Greenland due to its resolution of 30 km. The use of higher resolution data sets or in-situ measurements might unravel potential relations so far hidden. On the other hand, a persistent lack of evidence for external forcing might suggest IMW dynamics are mostly driven by variations in the internal state of the ice mélange.

### 5.3 Drivers of IMW episodes

To understand the main drivers for IMW episodes we employed a stochastic model of iceberg motion in a fjord. With the help of BRIMM, our simple 1D iceberg dynamics model, we were able to reproduce IMW episodes that are similar to those observed on satellite imagery. Obviously, the aim of investigations with such a simple model is not the realistic reproduction of real-world events. Rather, BRIMM facilitates the investigation of the relative importance of different processes, and is used for an assessment of the sensibility to different choices of model parameters, the length and time scales, and the emerging
dynamics. BRIMM produces a wide variety of responses, mainly depending on the magnitude, time interval and bias of the random iceberg motion.

In the BRIMM simulations, the amount of random motion in the fjord depends on the number of time steps and on the distance of random iceberg motion $\Delta x_{\mathrm{max}}$. In addition, the biased preferential motion (due to $p_b$) away from the glacier terminus dictates the flux of icebergs through the fjord, and therefore strongly influences the density of floating icebergs.

An oscillating, but long-term stable terminus positions is only occurring in a limited range of model parameters. To achieve this dynamical equilibrium, a prerequisite is that the rate of iceberg release from the glacier and the rate of iceberg transport by the biased random walk are of similar magnitude.





If the iceberg release to the fjord is too small (calving events of small size) and the fjord currents ($p_b$ in the model) rapidly carry away the icebergs, no dense ice mélange can develop. Without dense ice mélange, calving is occurring continuously
and its rate is fully determined by processes at the calving front. In deep water, the ice front rapidly recedes until it reaches a shallower pinning point (not used in the model runs shown here). Such a setting applies to many Greenland fjords, and has been well documented in the bay in front of Eqip Sermia (e.g. Walter et al., 2020; Wehrlé et al., 2021).

If, on the other hand, iceberg release to the fjord is too high (many calving events, large iceberg volumes released), paired with a low mobility of floating icebergs, the fjord becomes densely packed, and newly calved icebergs and glacier advance
preclude the formation of open water leads. Such packed ice mélange slows down IMW episodes which are often stopped halfway up the fjord, and therefore suppresses calving. In such a setting, the ice mélange occupies large parts of the fjord, and the glacier advances at the flow speed through the fjord until it reaches the fjord mouth with open water. Such large glacier advances are currently rarely observed in Greenland.

## 5.4   Time scales

Several time scales dominate the ice mélange dynamics. The one important assumption – implemented in the BRIMM model – is that calving is only possible if a narrow open water lead reaches the glacier terminus, or if a wide open lead forms at a certain distance (5 km in our model runs) from the terminus. We assume that the time scales intrinsic to the glacier which control calving are much faster. We therefore assume that the glacier is always ready to calve, as soon as an open water lead forms in vicinity of the calving front.

Under these assumptions, the only time scale dictated by glacier dynamics is the rate of terminus advance, i.e. the glacier speed. Its effect on the ice mélange is restricted to pushing floating icebergs ahead of the calving front, therefore forming a dense proglacial ice mélange that precludes calving. The rate of glacier motion in large outlet glaciers in Greenland is between 5 and 40 m/d. This means that the extent of a typical iceberg (100 m in our model runs) is covered within 3 to 20 days. In our model runs we chose a terminus advance speed of 15 m/d.

The other important time scale is related to the random motion of the ice mélange. It is given by the frequency with which icebergs move a random distance in a random direction (up- or downfjord), which corresponds to a time step in BRIMM. Based on arguments given below, we assume in our model runs that an iceberg moves 50 times a day according to Equation (1). This randomness is biased away from the glacier by $p_b$, representing ocean currents and wind drift in the real world.

Icebergs move mostly by seemingly random motion, driven by currents within the fjord and by wind forcing (FitzMaurice
et al., 2016). Strong tidal currents exert important drag forces on icebergs (Hughes, 2022) and drive them back and forth twice daily . The alternating flow of tides every 6 hours gives a good upper limit of the time scale for the random motion (i.e. 0.25 d). Fjord seiches, i.e. long-period waves within the fjord, move at wave speeds such that they have a recurrence time of about 30 minutes (i.e. 0.02 d; Amundson et al. (2008)).

An alternative line of reasoning considers how long it takes to accelerate and to stop an iceberg of 100 m length (see
Appendix A). Given the viscosity of water and the drag coefficient, a characteristic length scale $\Lambda$ of roughly the size of the iceberg (i.e. 100 m) is obtained. This means that the stopping distance of a moving iceberg is about 2.3 $\Lambda$. Assuming the



iceberg initially moves at 1 m/s, the stopping time scale is 100 s. The acceleration of an iceberg will be of similar magnitude given that the accelerating force has to be at least twice the drag force in water. Therefore the acceleration of an iceberg and the deceleration need some 500-1000 s. This simple argument shows that during a day there cannot be more than 100 random

motions. Again, a iceberg motion time step of 0.01 - 0.02 d emerges.

Based on the above arguments, we used a time step size of 1/50 d = 0.02 d for the BRIMM model runs, i.e. each iceberg moves twice per hour in a random direction by a random distance.

The ice mélange dynamics generated by BRIMM are not matching the observed behavior in much detail. But by changing a few more parameters, a much more chaotic response could be achieved. For example the number of calving blocks could be

varied either randomly, or dependent on proglacial ice mélange thickness.

## 6   Conclusions

The analysis of spaceborne observations in combination with model results suggest that the IMW process is an important control on the calving activity of KG, and to a lesser extent at HG and JI. While the dense ice mélange cover at the front of the glacier efficiently inhibits calving during the early stage of an IMW episode, the final stage of such an episode triggers

large-scale calving events. Results from a numerical model suggest that the observed cyclic IMW process is self-sustained and controlled by an IMW – calving feedback. Through this feedback, late-stage IMW episodes trigger new calving events, while the calving promotes the compaction and eventual yield of the ice mélange, thus giving rise to new IMW episodes.

An important conclusion from the modeling study is that slightly biased random motion of icebergs is sufficient to explain observed IMW dynamics. No fluctuating external forcing is needed to explain the cyclic behavior, the progress of IMW

episodes or the observed IMW propagation speeds. The observed ice mélange dynamics can be explained by random motion, with an emergent behavior of self-sustained punctuated dynamics, which is reminiscent of self-organized criticality (SOC; e.g. Jensen (1998)).

This study demonstrates the importance of short-term ice mélange dynamics on the calving activity of large Greenland outlet glaciers. While radar satellite imagery provides an almost daily revisit time over the entire ice sheet, the scale of the patterns it

can resolve — both spatially and temporally — remains limited compared to high-resolution field acquisitions that, however, are often of short duration. More and longer in-situ measurements are therefore needed to bridge this observational gap.

The study also underlines the importance of properly understanding the dynamics of floating icebergs, and especially their disintegration and melting due to heat advected by ocean currents in a fjord. Observing ice mélange conditions is very difficult, but is a prerequisite for predicting the future evolution of ice mélange dynamics around Greenland in the context of climate

change. Newly-established states will likely result in a strong and tight competition between processes affecting the cohesion of dense ice mélange, such as enhanced surface and submarine melt due to higher temperatures, and those promoting its spread and strengthening, that is mainly an intensification of fjord jamming due to a higher solid ice discharge.



A better understanding of such a complex, dynamic and heterogeneous environment can therefore only be achieved through a combination of different complementary observational and numerical approaches. Such an effort will eventually help resolving

the influence of current and future ice mélange dynamics on the longer-term stability of Greenland outlet glaciers.

*Code availability.* The version of the earthspy Python package used to process and download Sentinel-1 and Sentinel-2 images is presented in a Zenodo repository [in preparation]. The latest earthspy version can be installed from https://github.com/AdrienWehrle/earthspy. The BRIMM model is available at https://github.com/MartinLuethi/BRIMM.

*Data availability.* Sentinel-1 and Sentinel-2 satellite data are freely available on the ESA Open Access Hub (ESA, 2022). All IMW fronts

analyzed in this study have been publicly shared as shapefiles in a Zenodo repository: [in preparation].

*Supplement.* S1 to S4: Animations of Sentinel-1 HH images at KG showing the migration of IMW fronts (red lines) for the June-November period from 2018 to 2021. The overlaying red lines were removed on the left panels to fully appreciate the discontinuity between jam-packed and weak ice mélange. S5 to S7: Animations of BRIMM model results obtained with the parameter values listed in Table 1, 20 calving blocks, a length of terminus retreat after calving of 400 meters, a random walk bias of 0.02 and a maximum random walk step length

of 20, 100 and 150 meters for S5, S6 and S7, respectively. Light blue, dark blue and white areas represents the glacier, open water and ice mélange, respectively. The red line shows the position of the open water lead that is the closest to the glacier terminus. The y dimension has been expanded for visualization purposes only.

*Author contributions.* AW initiated the study with support from ML and performed the analysis of satellites images. ML developed and analyzed the BRIMM model. AW and ML drafted the manuscript. All authors contributed to the editing and reviewing of the manuscript.

All authors have read and agreed to the published version of the paper.

*Competing interests.* The authors declare that they have no conflict of interest.





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
