# Peer review of "The control of short-term ice mélange weakening episodes on calving activity at major Greenland outlet glaciers"

_EGUsphere, 2022_

## Author Comment (AC2)

**Response to reviewers: The control of short-term ice mélange weakening episodes on calving activity at major Greenland outlet glaciers**

Adrien Wehrlé, Martin P. Lüthi, Andreas Vieli

December 2022

We are grateful for the thorough and precise comments made by the two referees, Suzanne Bevan (Referee 1) and Surui Xie (Referee 2), that greatly helped us improving our manuscript. In the following, we address general and detailed comments point by point starting with Referee 1 and proceeding with Referee 2. Answers are written in blue and associated modifications are written in red in the revised manuscript and described here in the same color.

**1  Referee 1**

Line 22: delete 'with'. **We included this modification.**

Line 27: replace 'combination between' with 'combination with'. **We included this modification.**

Line 35: 'remains' **We included this modification.**

Line 58: replace 'brought' with 'provided'. **We included this modification.**

Line 61: replace 'The latter' with 'These studies' **We included this modification.**

Line 74: there is a random 'x' before '(Luckman'. **We included this modification.**

Line 92: replace 'Center' with 'central' **We included this modification.**

Line 109: replace 'Associated to' with 'Associated with'. **We included this modification.**

Line 121: replace 'quantified' with 'assigned'. **We included this modification.**

Line 152: where does the time scale needed for acceleration of a large iceberg derive from? I see later from Appendix A but there doesn't seem to be any Appendix available. **Appendix A indeed got lost in the initial manuscript submission. We added it in the revised manuscript.**

Line 158: 'implemented' **We included this modification.**

Line 167: 'turn over' would be better. **We included this modification.**

Table 1: The min and max values for Dxmax differ from those in the text (line 153) **We corrected those values in the text.**

Figs. 2-4: Add full dates to the panels, especially in fig. 4 where there are two events in 2018. **We decided to keep the month-day format to avoid having a lot of identical text that could distract the reader from the main message of the figure, i.e. the variations in propagation speeds through time (and hence the curve shapes more than the actual dates). The year is specified in bold for each subplot and we don't think it needs to be repeated next to each each point.**

Line 266: replace 'episodes than during' with 'episodes as during'. **We included this modification.**

Supplementary videos S5-S7 need a little text to explain them. Does the red bar mark the IMW front? If so, why is it so far up-fjord from the more broken mélange? **The different elements present on the supplementary materials (e.g. the red lines in videos S5-S7) are presented at the end of the manuscript in the "Video supplement" section. The IMW front is the limit between dense and weak ice mélange, which is very often further upfjord compared to the boundary between weak ice mélange and open water. We added more description about the surface visible on these videos in the caption of the supplementary materials L514-515.**

Fig. 6: Are the extrema in the pdfs determined by the actual observations or are they probability intervals? **The extrema of the violin plots correspond to the extrema of the associated distribution, as mentioned in the caption: "The central tick corresponds to the median, and the upper and lower ones to the distribution's extrema".**

Fig. 9: Change the axis titles to agree with the caption. **We included this modification.**

Line 356: this sentence needs rewriting somehow, it is not clear at the moment what is intended. **We rephrased this sentence.**

Line 361: I think it could be helpful here to expand on 'slow'. Robel (2017) talks about a few days, which fits in nicely with the minimum time observed between successive IMW propagation events. **We added a quantification for "slow" propagation which is of the order of several hours, only the return to background state is of the order of several days (also included).**

Line 433: 'faster than' what? **We mean faster than ice mélange weakening. We included this clarification.**

**2 Referee 2**

**3 General comments**

**3.1 Observations**

The observed timing and transition of IMW to iceberg calving process do not appear to follow a certain pattern. Apart from the description of some IMW episodes, the process can hardly be replicated in other fjords, nor in the BRIMM model described in this manuscript. Do we have sufficient data to quantitatively characterizing the IMW and calving processes and their possible feedback? If not, what are needed? **At Kangerlussuaq glacier, the IMW process is almost continuous during the period from June to November. We agree -and discuss in the manuscript- this process doesn't occur in most of Greenland's fjords because specific conditions (the combination of a high solid ice discharge and a narrow outlet) must be met. We believe more in-situ observations and monitoring of IMW episodes such as reported in Xie et al, 2019 are needed to better understand possible feedbacks. We also believe the data acquired by the two Sentinel-1 satellites is sufficient to draw a general, first characterization of IMW episodes but can't resolve in details the transition from IMW to calving. These different points were initially addressed in the "Conclusions" section, and has been clarified at L203-205.**

The IMW front propagation speed (Figure 6c and in the manuscript text) is probably not a useful variable in describing the short-term IMW episodes, as mélange break-up or collapse often behave like transient events. **We agree with this comment, supported by observations made by Xie et al 2019. However, at the event scale, IMW fronts at the front of Kangerlussuaq glacier do propagate upfjord over a median distance of 5.9 km and for 17 days and we therefore still find relevant to compute a propagation speed, a variable which is particularly interesting when e.g. compared with fjord geometry.**

**3.2 Modelling**

Shouldn't the bias added to the model be equal or similar to the advance rate of the glacier terminus? **Glacier motion and ice mélange biased displacement are two independent processes, the latter we think being driven by winds or ocean currents. The total ice mélange motion is the sum of the two displacements since the glacier pushes the floating ice downfjord. We don't see a reason for the mélange motion induced by glacier motion and ocean currents/winds to have equal or similar magnitudes.**

Meltwater plume or iceberg break-up can perhaps produce the two criteria implemented in the model, though calving is not necessarily to happen during or shortly after plume or iceberg break-up events. **We agree with this comment. We suggest the bias in the random walk we prescribe to potentially be linked to ocean currents and specifically to outgoing subglacial discharge through plumes below the floating ice cover. We clarified this suggestion at L149-151.**

**4 Detailed comments**

Line 35: remain → remains **We included this modification.**

Lines 47-48: This could confuse readers. Large calving events are more likely to occur when the mélange is unjammed, or the mass/extent of the jammed mélange has decreased to a critical level. **We clarified this sentence.**

Lines 86, 87 and a few other places: It is perhaps better to explain the terminology "solid ice" when first referred – if the authors would like to keep the term – I thought ice could only be solid in the nature environment. **We decided to remove the term "solid" used by Mankoff et al, 2021 to avoid confusion.**

Lines 78-179: Does "constant" mean fixed-coordinate in Eulerian frame? **The reference point is strictly fixed in space. We clarified this point.**

Lines 208-209: It may be true that the mélange strength and density contribute more at HG than at KG on some aspects of the mélange dynamics (in a relative sense), but how could the comparison of HG and KG's fjord geometry lead to a conclusion that mélange's strength density are more important than fjord geometry in affecting mélange dynamics? **We agree there is no clear evidence for the second point discussed by Referee 2. We modified this sentence to only note the first point here mentioned.**

Line 322: Red → Black **We included this modification.**

Lines 324-325: In figure 9a, under what circumstances would the terminus be stable? When the average advance rate of the calving front is 0 (white color)? Did the authors implied that the average advance rate of calving front needed to be a large negative value (towards -20 m/d) for the glacier to have a stable terminus position – "either the random motion or the bias has to be large"? **We consider the terminus to be at a stable position when the average advance rate of the calving front is close to 0, and added this clarification. We agree the**

**description of the relation between the two variables could be misleading and clarified it at L337-339.**

Line 326: What are the conclusions, could the authors elaborate? **As the conclusions for figures 9b and 9c are similar than for 9a, we resolved this comment through the precedent comment made by the referee.**

Line 352: How about ice blocks with high length-to-height ratio? I don't feel they will increase the mélange by smaller areas. **Blocks with high length-to-height ratios most of the time won't capsize at flotation and therefore won't increase their area at the fjord's surface compared to the one they were covering at the glacier's surface. We added this statement for clarification at L365-367.**

Lines 356-361: The first sentence stated that calving could increase the density of mélange, whereas the latter example suggests that calving can break the cohesion and perhaps reduce the density. They seem to be controversial. Some rewording is needed. **Robel et al 2017 showed that the propagation of a compression wave was observed in sea ice after a calving event and resulted in the creation of crevasses. This process is described in the sentence following the one here pointed by the referee. Nevertheless, linked to a similar comment from Referee 1 asking for clarity on the same statement, we rephrased this sentence.**

Line 389: HH → HG. **We included this modification.**

Line 418: So, the bias implanted in the model was caused by ocean currents? Please clarify this in the model set-up section. **We followed this advice following the first comment on the bias in the "Modelling" section of Referee's 2 review. See comment above and associated answer.**

Line 427: It is perhaps difficult, if not impossible for the glacier terminus to reach the fjord mouth. Under the scenario suggested by the authors, pro-glacial mélange should have some length to suppress calving and allow glacier advances. **We agree the scenario where the glacier front reaches the fjord mouth can't include an extensive pro-glacial ice mélange cover that would inhibit calving and hence glacier retreat. We modified for the scenario where the glacier would only approach the fjord mouth without reaching it (L447-448).**

Line 450: Where is Appendix A? **Appendix A indeed got lost in the initial manuscript submission. We added it in the revised manuscript.**

Figure 2: Some details about the IMW front detection should be provided in the caption or in the text of the manuscript. For example, the extent of mélange in panels a and b appears to be beyond the range of the colored lines, and the SAR images don't show significant contrast between strong weak mélange (particularly for panel b). Besides that, it would be good to add the reference line (for computing distances) onto the maps – for this and other similar figures. **We specify in the Method section of the manuscript and throughout the results that we are not mapping the extent of the ice mélange but the boundary between densely packed and weaker ice mélange. On this figure, we only show Sentinel-1 HH images. Sometimes the boundary between dense and weak ice mélange is only visible in the HV polarized backscatter as the latter is better at detecting changes in surface properties. The HH and HV modes have always been combined to detect the IMW front. We added this precision at L179-181.**

Figure 5: There are a few jumps/discontinuities in IMW distances to the reference point without calving activities (e.g., in late June and early October 2019, and late November 2020). Please provide some details about these jumps, otherwise it is difficult for the readers to interpret the figure. **We describe these events at L259-261. However, we realized we only mentioned the two events in June and October 2019, we added the third event in late November 2020 at L267-269.**

Figure 6 caption, first sentence: "at KG, HG, and JI"? **We first analyzed three IMW episodes per glacier at KG, HG and JI, then we detected 29 IMW episodes and KG. We clarified this point in the figure caption.**